# Mechanical Characterization of Human Fascia Lata: Uniaxial Tensile Tests from Fresh-Frozen Cadaver Samples and Constitutive Modelling

**DOI:** 10.3390/bioengineering10020226

**Published:** 2023-02-07

**Authors:** Lorenza Bonaldi, Alice Berardo, Carmelo Pirri, Carla Stecco, Emanuele Luigi Carniel, Chiara Giulia Fontanella

**Affiliations:** 1Department of Civil, Environmental and Architectural Engineering, University of Padova, 35131 Padova, Italy; 2Department of Biomedical Sciences, University of Padova, 35131 Padova, Italy; 3Centre for Mechanics of Biological Materials, University of Padova, 35131 Padova, Italy; 4Department of Neuroscience, Institute of Human Anatomy, University of Padova, 35121 Padova, Italy; 5Department of Industrial Engineering, University of Padova, 35131 Padova, Italy

**Keywords:** Fascia Lata, uniaxial tensile test, biomechanical characterization, constitutive modelling

## Abstract

Human Fascia Lata (FL) is a connective tissue with a multilayered organization also known as aponeurotic fascia. FL biomechanics is influenced by its composite structure formed by fibrous layers (usually two) separated by loose connective tissue. In each layer, most of the collagen fibers run parallel in a distinct direction (with an interlayer angle that usually ranges from 75–80°), mirroring the fascia’s ability to adapt and withstand specific tensile loads. Although FL is a key structure in several musculoskeletal dysfunctions and in tissue engineering, literature still lacks the evidence that proves tissue anisotropy according to predominant collagen fiber directions. For this purpose, this work aims to analyze the biomechanical properties of ex-vivo FL (collected from fresh-frozen human donors) by performing uniaxial tensile tests in order to highlight any differences with respect to loading directions. The experimental outcomes showed a strong anisotropic behavior in accordance with principal collagen fibers directions, which characterize the composite structure. These findings have been implemented to propose a first constitutive model able to mimic the intra- and interlayer interactions. Both approaches could potentially support surgeons in daily practices (such as graft preparation and placement), engineers during in silico simulation, and physiotherapists during musculoskeletal rehabilitation, to customize a medical intervention based on each specific patient and clinical condition.

## 1. Introduction

Human Fascia Lata (FL) is an aponeurotic fascia, a type of deep muscular fascia, which covers the thigh muscles. In other words, it is a dense and regular connective tissue with a multilayered organization. In fact, the biomechanics of FL is influenced by its arrangement in several layers (usually two) separated by loose connective tissue. The latter, rich in hyaluronic acid which allows gliding during muscles’ contractions, is present between the layers of this aponeurotic fascia, and between these layers and the muscle epimysium. In terms of thickness, the FL is composed of layers of 277 µm (±86 µm) each, separated by layers of loose connective tissue of 43 µm (±12 µm) each, leading to a composite structure with an average total thickness of 944 µm (±156 µm) [1].

Pursuant to the concept of myofascial continuity, FL could be seen as a large flat tendon that receives and transmits muscle forces through myofascial expansions. Therefore, its fibers, mainly of collagen type I (≈80%), are arranged according to in vivo loadings, thus defining the biomechanical properties of FL. In both layers, most of the collagen fibers run parallel in a distinct direction (interlayer angle: 75–80°, Figure 1a), mirroring the fascia’s ability to adapt and withstand specific tensile loads [1].

In the past years, FL has attracted the interest of the scientific community, thanks to its physiological and structural functions, combined with a relative non-cellularity and low nutritional requirements, which fit the required characteristics for grafting applications [2,3,4].

Currently, the in vivo study of the biomechanical properties of the fibers of the FL is difficult in terms of their accessibility. For this reason, to date, ex vivo experimental analysis has been the most frequent choice [5,6,7,8]. From a mechanical point of view, the FL has been characterized especially by uniaxial failure tests, but only a few works reported quantitative information related to strength [9,10,11], strain at break [4,9], and elastic parameters such as Young’s modulus [12,13]. In addition, many factors may influence the interpretation of the results, such as a balanced sample size; sex and age of the donors [5]; precision in the harvesting site [5,12]; tissue conservation/thawing/hydration/testing [6,14,15]; sample preparation and measurements [16]; setup characteristics [9,17]; and testing protocols with preconditioning cycles [10,18].

Indeed, even if the collagen bundles’ directionality, which characterizes the connective layers, is of particular relevance to FL biomechanics, to date the literature is still poor of a clear description of FL behavior as an anisotropic, viscoelastic, fiber-reinforced composite material in order to highlight possible differences between layers contribution.

For these reasons, we investigated this open theme as the main purpose of this research work, through experimental evidence and constitutive modelling, to quantify mechanical differences with respect to fiber directions within the FL composite structure.

In particular, we performed mechanical tests on FL samples obtained from fresh-frozen cadavers and developed a constitutive model able to replicate each layer’s contribution to the overall FL behavior. These results could support clinical practices, for example guiding surgeons during graft preparation and placement; physicians in risk assessment after surgery; physiotherapists during musculoskeletal rehabilitation (manual therapy); and engineers for in silico modeling towards personalized medicine.

## 2. Materials and Methods

### 2.1. Human Sample Collection

FL was harvested from the anterior compartment of the thigh, 10 cm from the anterior superior iliac spine along the line connecting this latter to the upper border of the patella. FL patches were wrapped in plastic bags and stored (−80 °C) from four fresh-frozen human donors (2 females and 2 males, aged 54–89 y/o, no pathologies or previous surgeries were reported in their clinical history, duration of the preservation less than one year, first thawing cycle), in agreement with the *Body Donation Program* of the Institute of Anatomy of the University of Padua [19]. Tests were performed between November 2021 and February 2022; samples were thawed at room temperature and tested within 12 h, after being hydrated and cut into rectangular strips as composite (i.e., two connective layers separated by loose connective tissue).

The multiple samples were cut using a preformed razor (available from the mechanical tester manufacturer) on a cutting board, oriented and loaded during testing according to the directions of the main collagen bundles for each individual layer, principal and transverse (namely L1 and T1, the longitudinal and transversal fiber distribution of layer 1, and L2 and T2 for the second layer, see Figure 1). The sample size for each configuration was defined based on the availability of native tissue for each subject (patches of approximately 12 cm × 8 cm each). All the samples were kept moist during and after the cutting from the specimen with phosphate-buffered saline solution to not dry or shrink throughout the entire testing procedure.

### 2.2. Experimental Tests on Human Samples

Sample temperature was monitored using an infrared scan prior to testing (for each sample, values from 19–21 °C). The grip-to-grip length and width for all specimens as well as the cross-sectional areas were calculated on the average values acquired with a manual caliper (resolution: 0.1 mm) in several points of the samples (Figure 2a). The average interlayer angle for each patch was computed using image processing techniques through ImageJ software [20]. Mean values were respectively for: initial length = 30 ± 1 mm, thickness = 0.68 ± 0.14 mm, width = 3.70 ± 0.43 mm. As schematically shown in Figure 2, a rigid “C”-shaped support was made to fix the ends of the samples (the portions in contact with the grips) with a commercial superglue. To avoid slippage, the extremities were folded twice at the ends of the holder. The C-shape allows the specimen to be kept under a small constant tension to avoid bending during the dimensional measurements and assembly into the testing machine (avoiding length reduction and assuring the correct tensioning). The specimens were aligned into the gripping mechanism (serrated grips with rough surfaces to be in contact with soft tissues and prevent slippage). All the specimens were first mounted on the upper handle to allow their alignment by gravity. To avoid tissue damage or stress concentration, the grip-specimen-grip force at their interface (prior to cutting the rigid support and testing, Figure 2b,c) was also checked. Samples were kept moist during overall testing by pipetting them with phosphate-buffered saline solution, to prevent possible changes due to dehydration.

The mechanical tester that was used for the experimental campaign is a Model Mach-1, ^©^Biomomentum Inc., Laval, QC, Canada, with a load cell capacity of 250 N and accuracy of ±0.0125 N. The data were recorded with a sample rate of 10 Hz. Ten cycles (according to values reported by Gordon et al., 2017 [7]) of preconditioning, with a strain rate of 1% s^−1^, were performed prior to each test, to achieve a repeatable and stable response with a small under-breaking tensile load with a cyclic tensile test [21]. Then 15 s of resting time was observed before failure or stress-relaxation protocol. Failure was achieved with a strain rate of 0.5% s^−1^. Stress relaxation was performed by applying 5 strain incremental ramps (2% each, 1% s^−1^) and maintaining each ramp for 300 s, similarly to previous studies [14].

### 2.3. Constitutive Modelling of Elastic Behavior

The structural configuration of FL, such as a multi-layered anisotropic system, and results from mechanical experimentations, which revealed moderately large strain phenomena and non-linear elasticity, suggested describing the elastic stage of the mechanical behavior in the framework of fiber-reinforced materials hyperelasticity. Considering the micro-structural arrangement of the two layers that mainly compose the FL, a two-fiber families Holzapfel–Gasser–Ogden formulation was assumed to characterize the FL composite response [22]:(1)$$W\left(C\right)=K\left(\frac{J^{2}-1}{2}-\ln J\right)+C_{1}\left({\bar{I}}_{1}-3\right)+\overset{2}{\underset{\alpha =1}{\sum }}\frac{k_{1\alpha }}{2k_{2\alpha }}\left\{\exp\left[k_{2\alpha }{\left\langle \bar{E_{\alpha }}\right\rangle }^{2}\right]-1\right\}$$
(2)$${\bar{E}}_{\alpha }=\kappa_{\alpha }\left({\bar{I}}_{1}-3\right)+\left(1-3\kappa_{\alpha }\right)\left({\bar{I}}_{4\alpha }-1\right)$$
where *W* is the strain energy function, *J* is the deformation Jacobian, such as J=det F and **F** is the deformation gradient, ${\bar{I}}_{1}$ is the first invariant of the isovolumetric part of the right Cauchy–Green strain tensor $\bar{C}=J^{-2/3}F^{T}F$. ${\bar{E}}_{\alpha }$ describes the *α*th fiber family isovolumetric strain considering the spatial distribution of the fibers. ${\bar{I}}_{4\alpha }$ evaluates the square of tissue isovolumetric stretch along the *α*th fiber family preferential direction $a_{0\alpha }$. The spatial distribution of fibers around the direction $a_{0\alpha }$ is defined by means of the parameter $\kappa_{\alpha }$: if $\kappa_{\alpha }=0$ fibers are perfectly aligned along direction $a_{0\alpha }$, if $\kappa_{\alpha }=1/3$ fibers are randomly distributed in space. The operator ⟨·⟩ stands for the Macauley bracket and is defined as $\langle x\rangle =\frac{1}{2}\left(x+\left|x\right|\right)$. *K* and *C*_1_ are isotropic parameters that specify the initial volumetric and shear stiffness of the tissue, respectively. $k_{1\alpha }$ evaluates the fibers initial stiffness, while $k_{2\alpha }$ describes fiber stiffness evolution with strain (i.e., because of uncrimping phenomena).

The identification of constitutive parameters was performed by means of the inverse analysis of tensile tests. The action required to compute the nominal stress tensor [23]:(3)$$\begin{array}{c} P=K\left(J^{2}-1\right)F^{-T}+2J^{-\frac{2}{3}}(C_{1}1-\frac{1}{3}I_{1}F^{-T})+\\ +\sum_{\alpha =1}^{2}\begin{array}{c} k_{1\alpha }\exp\left[k_{2\alpha }{\left\langle \bar{E_{\alpha }}\right\rangle }^{2}\right]\left(\frac{\left|\bar{E_{\alpha }}\right|}{\bar{E_{\alpha }}}+1\right)\cdot \\ \cdot \left[\kappa_{\alpha }J^{-\frac{2}{3}}\left(1-\frac{1}{3}I_{1}F^{-T}\right)+\left(1-3\kappa_{\alpha }\right)J^{-\frac{2}{3}}\left(I_{4\alpha }{}^{\frac{1}{2}}a_{\alpha }⊗a_{0\alpha }-\frac{1}{3}I_{4\alpha }F^{-T}\right)\right] \end{array} \end{array}$$
where $I_{1}$ is the first invariant of the right Cauchy–Green strain tensor $C=F^{T}F$, $I_{4\alpha }$ evaluates the square of tissue stretch along the *α*th fiber family preferential direction $a_{0\alpha }$, while $a_{\alpha }$ specifies the *α*th fiber family preferential direction in the deformed configuration.

The predictive model of a tensile test along direction **e***_i_* is developed considering the stress component P*_ii_* and imposing null values to stress components along the orthogonal directions **e***_j_* and **e***_k_*, such as P*_jj_* = P*_kk_* = 0. A cost function Ω evaluates the discrepancy between model results P*_ii_* and experimental data depending on the assumed set of constitutive parameters ω:(4)$$\Omega \left(\omega \right)=\sum_{j=2}^{n}\frac{1}{n}{\left\{2-\frac{P_{ii}{}^{mod}\left(\omega ,C^{r}\right)}{P_{ii}{}^{exp}\left(C^{r}\right)}-\frac{P_{ii}{}^{exp}\left(C^{r}\right)}{P_{ii}{}^{mod}\left(\omega ,C^{r}\right)}\right\}}^{2}$$
where *n* is the number of experimental data, $P_{ii}{}^{mod}\left(\omega ,C^{r}\right)$ is the model prediction at the *r*th experimental strain condition $C^{r}$, $P_{ii}{}^{exp}\left(C^{r}\right)$ is the corresponding experimental result. The proposed cost function is assumed to compute the discrepancy between experimental and model results by including data from all the tensile tests at disposal, such as the ones performed along the two main directions of fiber families and along the two orthogonal directions to fiber families. Because of the adopted hyperelastic constitutive formulation, the analysis must account for experimental data in the elastic region only, which extended up to 3–4% strain condition.

The minimization of the cost function, by means of coupled stochastic and deterministic optimization algorithms, led to the optimal parameters. Aiming to ensure the thermodynamic consistency of parameters, positive values of tangent Young modulus $E_{i}{}^{T}$ and tangent Poisson ratios $\nu_{ij}{}^{T}$ and $\nu_{ik}{}^{T}$ were imposed by introducing penalty terms into the cost function [23]:(5)$$E_{i}{}^{T}=\lambda_{i}\frac{\partial \sigma_{ii}}{\partial \lambda_{i}}$$
(6)$$\nu_{ij}{}^{T}=-\frac{\lambda_{i}}{\lambda_{j}}\frac{\partial \lambda_{j}}{\partial \lambda_{i}}\;\;\;\nu_{ik}{}^{T}=-\frac{\lambda_{i}}{\lambda_{k}}\frac{\partial \lambda_{k}}{\partial \lambda_{i}}$$
where $\lambda_{i},$ $\lambda_{j}\;\mathrm{and}\;\lambda_{k}$ are stretch components along and normally to the uniaxial loading direction, while $\sigma_{ii}$ is the normal component of the Cauchy stress tensor along direction **e***_i_*, where $\sigma =\frac{1}{J}PF^{T}$.

### 2.4. Statistical Analysis

Outliers were discarded and results analysis was computed with MATLAB 2021b [24]. To evaluate the potential difference in tissue mechanical behavior according to different subjects, layers, and directions, a statistical analysis was performed using Minitab Statistical Software [25] in terms of descriptive and inferential statistics (hypothesis tests: paired sample two-tail *t*-test, level of significance 0.05).

## 3. Results

Tensile tests were performed on samples of FL to observe possible influences of fiber orientations in tissue anisotropy. Moreover, tissue variability among the subjects was quite evident, being a common factor that characterizes biological tissues.

From the resulting failure curves, strength, strain at break, and Young’s Modulus referring to the linear region for the different groups of specimens were calculated. The % of stress relaxation from stress-relaxation curves was obtained for both longitudinal and transversal directions of each layer.

In the following subsections, results from different tests and directions of the applied load are presented, followed by the constitutive interpretation.

### 3.1. Failure Tests

The uniaxial response of the FL exhibits the standard behavior of soft connective tissues, with an increasing stiffening well shown in the first part of the stress–strain curve (Figure 3), up to an almost linear relationship that lasts until the progressive breakage of the microstructure propagates and thus an increase in strain leads the tissue to failure.

Among subjects, the FL exhibits similar strain at break, independently from the direction of the applied load, with values varying from 8.4% to 23.2%, with an average value of 16.4%, close to results of other authors [2,9,13]. Ultimate tensile strength (UTS) spans values ranging from 0.5–12 MPa, with strong differences between directions. In particular, as reported in Figure 3, there is always one layer stiffer and tougher than the other. This aspect is also highlighted by the Young’s Modulus obtained from the linear part of the tensile curves (Table 1). It resulted in 4.5–28 times higher when applying a load in the longitudinal than the transverse direction for the stiffer layer, while trends reversed for the softer layer, as a result of the collagen fiber orientation on both layers.

### 3.2. Viscoelastic Behavior

The viscoelastic response of the tissue during relaxation tests was evaluated in terms of normalized stress (stress divided by the peak stress value at the beginning of the relaxation process) with respect to time.

In Figure 4, the normalized median stress curves in time are reported for all the subjects, with a confidence interval (50%).

No considerable differences were found for the stress decay between directions, layers, and subjects. For longitudinal and transversal samples (L1 and L2 vs. T1 and T2), the stress at 60 s was on average 57% and 55% of the peak stress, respectively, thus the difference in the decay after 60 s between directions could not be considered statistically significant (*p* = 0.34). At the end of the relaxation time (300 s), it reached 44% of the peak stress for the longitudinal and 42% for the transversal direction. Also in this case, the difference in decay between directions (*p* = 0.55) was not statistically significant. From these observations, on average, the FL residual stress is almost halved after 60 s, and reduced up to 57% of the initial stress after 300 s. All these results are reported in Table 2.

Normalized stress-relaxation curves were fitted with a two-term Prony series, following this equation:(7)σnorm=1−γ11−etτ1−γ21−etτ2

### 3.3. Hyper-Elastic Modelling

A two-fiber families Holzapfel–Gasser–Ogden formulation was assumed to interpret the elastic behavior of FL. The implementation of specific model formulations and optimization algorithms made it possible to identify constitutive parameters of the different subjects S1, S2, S3, and S4 (comparison between the models and the experiments are reported in Figure 5). Parameters are reported in Table 3.

Figure 5 reports the discrepancy (with Ω computed by means of Equation (4)), between model results and experimental data, with regard to the different subjects and the different loading directions.

## 4. Discussion

The experimental results obtained from the failure tests of human FL with samples oriented along the two principal directions of the collagen fibers (i.e., the two layers) highlighted a fundamental feature that had not been reported previously, which supports the hypothesis that the two parallel layers of connective tissue contribute differently with reference to tissue strength and consequent stiffness (Figure 3 and Table 1). The FL is a composite structure and, when subjected to a uniaxial displacement (such as the tensile test), the two parallel layers deform likewise until the breakage, thus no differences were observed between the strains at break along the analyzed directions. This is not the case for the strength and thus the elastic modulus, since, for every subject, one layer was significantly stiffer than the other one, with reference to the overall response of the composite. This being the case, the total tensile load can be seen to be the contribution of all the layers, where the stiffest support the greatest load. In these terms, collagen fiber orientation with respect to the imposed displacement also plays a key role.

From these insights, we may suppose the two layers could be characterized by different thicknesses or collagen fiber density with direct reflexes on the strongly variable mechanical response. This behavior could also be motivated by the different functional roles played by the two layers, such as connecting the knee-hip axis or adapting to the muscular contraction and consequent volume variation.

Moreover, testing the FL as a composite means that the mechanical response of one layer will be a result of not only its collagen bundle direction, but also of the fiber distribution of the second layer depending on the above-mentioned interlayer angle. In particular, the more the average interlayer angle is close to 90° (i.e., the two families of fibers are almost orthogonal), the more the transversal response of one layer is close to the longitudinal response of the second one, with a clear example shown by subject 4 (Figure 3d), where results for L1 are almost mirrored on T2 and vice versa.

These observations are fundamental outcomes to consider when planning FL application for graft purposes: the direction along which a patch of FL will be harvested and implanted will impact the final mechanical behavior of the tissue when subjected to in vivo loads.

Referring to additional mechanical results obtained from the failure behavior, considerable differences were obtained among the subjects. As we stated in the introduction, there are a variety of factors that may influence the mechanical properties of biological tissues starting from the donor’s characteristics, such as age, sex and physical constitution. For this reason, we did not compare or report an average result among the subjects, but we focused on the differences within each subject. Future works could focus on the correlation between mechanical properties and subject variability (in terms of age, gender, clinical history, and so on). However, if we compare the overall mechanical quantities with the current literature, we found an experimental strain at break close to similar works (such as [9,10,13]), while on average UTS and Young’s modulus appear lower than some available results (e.g., [7,11,13]), but in agreement with [5,12]. There are several reasons that might be considered when comparing the results. First of all, the age of the subjects, which for [7,9,10,13] is minor than 60 y/o, while in the here-reported experiments, samples were harvested from 54–89 y/o donors. In addition, none of these authors highlighted the direction of the fibers along the specimens, while [5] analyzed tensile properties of samples oriented parallel or perpendicular to the fibers (even without considering FL multilayered structure arrangement) and our results rank in similar intervals. Other features could have influenced past or present results, such as the sample conservation (temperature, cycles of freezing and unfreezing), or the hydration methods (saline solution, pipetting or submerging the samples before and/or during the tests). Unfortunately, some information is not always available or reported, thus stressing the importance of always providing a clear and repeatable protocol.

When including the viscoelastic response of the FL for the four groups of tested samples, no significant differences were noticed (Figure 4 and Table 2) among the subjects and the layers; thus we can infer that no anisotropy affects the viscous properties of this connective tissue. About 50% of the stress relaxed after about 60 s, with another 10% of stress relaxation after the resting 240 s, resulting in a greater stress reduction with respect to other fascial tissues such as crural fascia [14]. Further tests are needed to characterize the equilibrium stress-strain curves for both layers and quantification of their viscous parameters.

To model the FL mechanical behavior, the constitutive analysis of FL tissues was performed aiming to mathematically compare experimental results from different subjects and from tests performed along different directions. With specific regard to constitutive parameters (Table 3), similar values, or at least similar orders of magnitude, characterized the parameters of the different subjects. Furthermore, acceptably low values of discrepancy between the model and experimental results were achieved (Figure 5). These outcomes reveal the consistency of the experimentations performed and the suitability of the adopted hyper-elastic formulation. The values of parameters $\kappa_{1}$ and $\kappa_{2}$ confirm the small/moderate dispersion of fibers along the two principal directions, which is visually observable on the samples (see as an example Figure 1b). Non-linearity parameters $k_{21}$ and $k_{22}$ suggest the strong stiffening behavior of FL, which agrees with the physiological function of the tissue.

It is necessary to also include the limitations of the experimental results and assumed constitutive analysis, which exclusively interprets the elastic stage of the stress-strain curves. Moreover, tests performed on biological samples are usually characterized by a non-negligible variability, and unfortunately, a limited number of samples were realized for every case study, due to material availability and patch dimensions. In addition, the freezing-thawing procedure to store and then use the samples could also affect the results. In this regard, in order to limit the number of freezing-thawing cycles, we reduced this bias by thawing, preparing, and testing the samples progressively during the laboratory sessions. The availability to perform only uniaxial tensile tests instead of biaxial tests could be seen as an additional limitation to our work, even if uniaxial tests were able to highlight the differences between tensile directions with respect to the principal fiber directions as well. Future studies should also include the analysis of fiber orientation changes in real time during the tensile procedure, since this still-missing aspect could better explain how the two connective layers interact when the tissue is stretched. With specific regard to the performed experiments, the elastic region ranges from 0% to 3–4% strain condition and within the assumed range, the uniaxial stress-strain curves exhibit a convex trend, which is typical of the elastic behavior of soft biological tissues. Stiffness loss and failure phenomena occur at higher strain conditions. More refined elasto-damage models should be adopted to interpret such damage processes, while visco-hyperelastic formulations are mandatory to analyze stress relaxation effects.

## 5. Conclusions

Experimental description of the mechanical behavior of human FL is a key feature in many clinical and tissue engineering applications. Some information is available from the literature, but nothing is known about the interlayer anisotropy and viscoelastic properties of such tissue. This study represents an experimental characterization of FL as a composite, in order to quantify layer contributions and influence of the fiber orientation on the overall mechanical response of the tissue. Experiments on human samples were carried out to identify failure and viscoelastic properties. A clear anisotropy was highlighted when comparing the elastic properties of the tissue with respect to fiber directions within the layers, while an almost isotropic response was observed concerning the viscous contribution. Finally, a first constitutive analysis was realized and validated with the experiments, providing useful insights on this fibrous connective tissue that is spreading in a variety of graft applications, in which a deeper knowledge of its mechanical behavior is needed for successful clinical results.

## Figures and Tables

**Figure 1 bioengineering-10-00226-f001:**
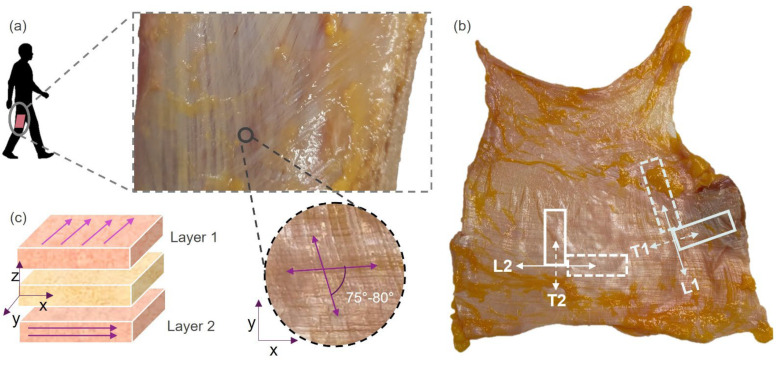
(**a**) Origin of the harvested specimens and histological evidence of the interlayer angle. (**b**) Example of harvested samples from one subject: numbers 1 or 2 refer to layers 1 or 2, while letters L or T refer to sample cutting longitudinal to the fibers or transversal to the fibers. Horizontal direction approximately corresponds to the cranio-caudal direction. (**c**) The multilayer structure and layer orientations.

**Figure 2 bioengineering-10-00226-f002:**
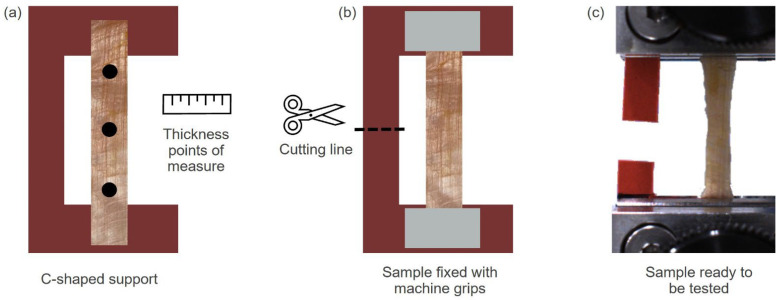
Sample preparation for mechanical testing. (**a**) Rectangular samples were positioned on rigid C-shaped supports and fixed with commercial superglue. Thickness was measured in at least three points (black dots) along the sample with a manual caliper. (**b**) Once the support with the sample was anchored to the machine grips, the support was cut, in order to not affect the mechanical test. (**c**) The sample is ready for the uniaxial test.

**Figure 3 bioengineering-10-00226-f003:**
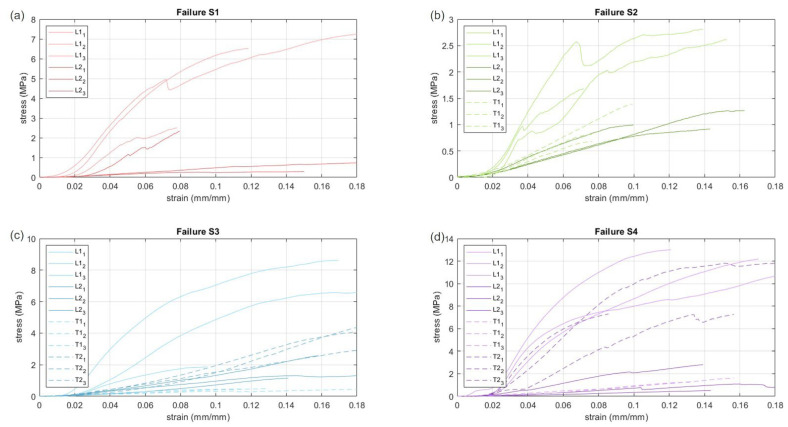
From (**a**–**d**) tensile tests up to failure for the fourth subjects along longitudinal (L) and transversal (T) directions of layer 1 or 2.

**Figure 4 bioengineering-10-00226-f004:**
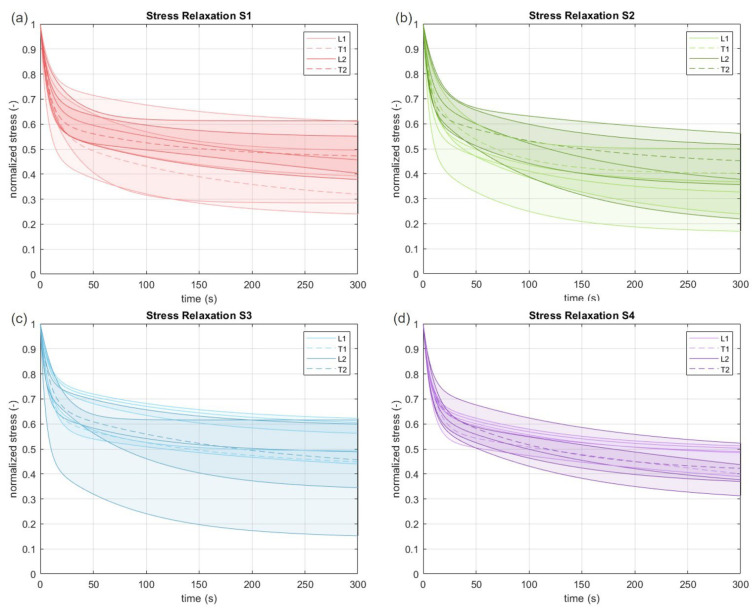
From (**a**–**d**) median stress relaxation curves after 300 s of relaxation time for the fourth subjects along longitudinal (L) and transversal (T) directions of layer 1 or 2. Color bands refer to the first and third percentiles.

**Figure 5 bioengineering-10-00226-f005:**
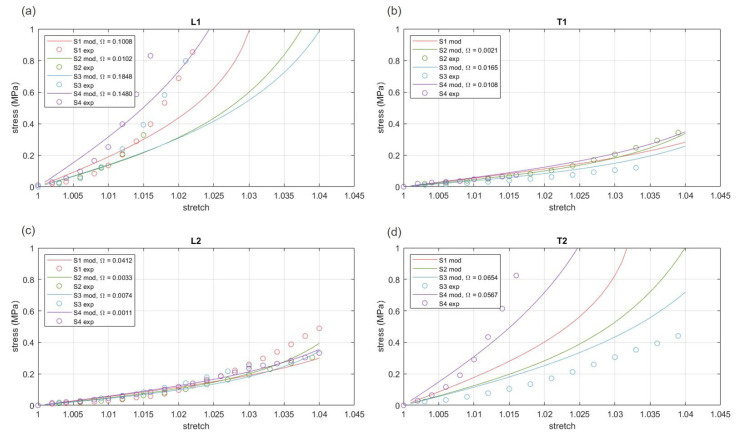
Evaluation of discrepancy between hyper-elastic model results and experimental data for the different subjects and the different loading directions (**a**–**d**). Discrepancy Ω was evaluated by means of the cost function reported by Equation (4).

**Table 1 bioengineering-10-00226-t001:** Ultimate Tensile Strength (UTS), strain at break and Young’s Modulus of FL samples subjected to tensile tests up to failure. Mean and standard deviation (SD) for each subject and testing direction.

Subject	Interlayer Angle (°)	Sample Orientation	UTS (MPa)		Strain at Break (% mm/mm)		Young’s Modulus ^1^ (MPa)	
			Mean	SD	Mean	SD	Mean	SD
S1	60	L1	5.4	2.7	13.2	6.0	94.3	23.1
		T1	N/A	N/A	N/A	N/A	N/A	N/A
		L2	0.8	0.4	22.2	12.5	18.9	24.1
		T2	N/A	N/A	N/A	N/A	N/A	N/A
S2	81	L1	2.1	0.7	12.1	4.3	55.7	7.7
		T1	1.0	0.4	8.4	8.4	12.5	1.6
		L2	1.1	0.2	13.5	3.2	10.8	3.7
		T2	N/A	N/A	N/A	N/A	N/A	N/A
S3	77	L1	5.2	4.8	13.2	5.3	66.9	34.9
		T1	0.5	0.0	23.2	19.1	4.5	1.1
		L2	2.0	0.7	20.0	8.8	11.0	4.3
		T2	4.1	1.1	19.8	1.4	13.8	3.5
S4	86	L1	12.0	1.0	16.3	3.8	191.9	54.5
		T1	1.4	0.2	14.9	1.5	6.9	2.0
		L2	1.5	1.2	18.0	6.7	10.0	9.7
		T2	9.0	3.0	18.2	11.1	119.9	65.8

^1^ referring to the linear region.

**Table 2 bioengineering-10-00226-t002:** Normalized residual stress (% of the peak, which is equal to 100%) after stress relaxation. Median is reported after 60 s and at the end of the relaxation time (300 s). Fitted parameters *γ* and *τ* refer to the two-term Prony series reported in Equation (7).

Subject	Sample Orientation	Residual Stress (%)		Fit Coefficients			
		After 60 s	After 300 s	*γ* _1_	*τ*_1_ (s)	*γ* _2_	*τ*_2_ (s)
S1	L1	51	39	0.41	6.37	0.22	124.38
	T1	48	32	0.43	10.04	0.29	158.14
	L2	59	46	0.35	8.31	0.25	199.53
	T2	55	47	0.38	7.70	0.15	112.39
S2	L1	46	33	0.43	6.27	0.25	105.29
	T1	52	40	0.27	6.33	0.33	57.54
	L2	58	38	0.29	7.19	0.38	150.48
	T2	57	45	0.35	6.11	0.22	134.94
S3	L1	70	61	0.24	8.06	0.18	132.55
	T1	56	45	0.37	7.67	0.21	140.28
	L2	57	49	0.35	5.44	0.17	94.57
	T2	60	46	0.33	8.45	0.27	182.56
S4	L1	60	50	0.32	6.65	0.19	114.18
	T1	53	40	0.42	9.71	0.46	615.57
	L2	54	38	0.36	7.88	0.32	178.57
	T2	57	42	0.31	7.02	0.28	108.78

**Table 3 bioengineering-10-00226-t003:** Constitutive parameters for the different subjects.

Subject	K (MPa)	$C_{1}\;(\mathrm{MPa})$	$k_{11}\;(\mathrm{MPa})$	$k_{21}$ (-)	$\kappa_{1}$ (-)	$k_{12}\;(\mathrm{MPa})$	$k_{22}$ (-)	$\kappa_{2}$ (-)
S1	0.0754	0.6807	2.7894	55.9368	0.0004	1.9971	578.9087	0.1879
S2	0.0163	0.3597	4.3575	173.3235	0.0767	0.7668	197.6555	0.0392
S3	0.0560	0.1071	5.7467	79.3530	0.1006	1.4018	141.6217	0.0746
S4	0.0431	0.4693	5.3137	42.1639	0.0004	0.9011	115.2489	0.0393

## Data Availability

The raw/processed data required to reproduce these findings cannot be shared at this time due to technical issues but are available upon direct request to the corresponding author.

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
