# Peer review of "Mechanical Characterization of Human Fascia Lata: Uniaxial Tensile Tests from Fresh-Frozen Cadaver Samples and Constitutive Modelling"

_bioengineering, 2023, doi:10.3390/bioengineering10020226_

Round 1

Reviewer 1 Report

General: The authors conducted a biomechanical study to test the direction dependent differences of fascia lata. The work is interesting and will contribute to soft tissue surgical or biomechanical simulation. However, the manuscript needs to be improved in several ways before its publication in this Journal can be granted. My concerns and specific comments are provided below.

Page 2 Line 53–61: Use of frozen cadaver seems to be a unique point of this study compared with the previous study (similar to fresh cadaver compared with those of formalin fixated). Please emphasize that in the introduction.

Page 2 Line 76–81: After the period of thawing, is there any morphological or mechanical change of the sample (e.g., getting dry or shrink)? The material to hydrate the sample before the test should also be explained in the method.

Page 3 Figure 1: Please put the information of the orientation of the fascial graft refer to the thigh into the figure b).

Page 3 Line 98: Please describe the actual number of the point that the author measured the thickness of the specimen.

Page 3 Line 101–102: It is difficult for the reader to imagine what the “rigid C shaped support” means. Please show it in the picture or graphic form in the Figure.

Page 4: The first formula (2) should be replaced to formula (1).

Page 9 Line 282–Page 10 Line 289: The elastic properties of the FL shown in table 1 and 3 seem to be different not only by the tensile direction and fiber angle but also the harvest site and each donor. Are there any other considerable factors to define the elastic characteristics of the FL, for example, sex, physical constitution, or underlying muscle size?

Author Response

We have addressed the reviewer's comment in the attached file. Thank you

Reviewer 2 Report

The study reports important experimental findings regarding the biomechanical properties of the fascia lata, notably the quantification of layer contributions and influence of the fiber orientation on mechanical behaviour of the tissue.

Overall, the manuscript is well-written, although the English language quality would require moderate revisions to improve the clarity of the presentation.

Specific comments are elucidated below:

Major comments

Lines 86, 200, 207, 225, 252, 256, 268, 287, 310, 322, 326 Error! Reference source not found. There appears to be a problem with the reference management system used in the manuscript. Several of the intext citations are not shown (instead the above error notice is displayed), hence the sources could not be accessed for the review. No specific comments can therefore be made in the contexts of the unknown citations in the text, and the validity of the reference claims could not be ascertained.

Line 76: No description is provided regarding the specific site of fascia tissue harvesting. Given the extensive nature fascia lata, with potential variations of mechanical properties and behaviour in different regions of the fascia, it is reasonable that harvesting should be done at a fixed site in all donors, for experimental consistency.

Line 77: Beside sex and age, no further information is provided on the donors regarding other factors that may affect the properties of the fascia such as duration of low-temperature preservation, body mass, history of diabetes, connective tissue disorders, previous surgeries, etc. This is in apparent contradiction of the authors’ stress on the importance of a clear and repeatable protocol (line 307)

The Discussion should also reflect some arguments regarding the choice of frozen bodies for this experiment, considering the known impact of freezing and thawing on the mechanical characteristics and behaviour of tissues.

Also, despite sample harvesting from 2 male and 2 female donors, no analysis or discussion is provided on findings with regard to sex and the implications thereof. Similarly, the potential impact of donor age or other known clinical characteristics on the biomechanical findings are not reflected in the discussion.

The limitations of uniaxial tensile tests in comparison to biaxial tests of directional biomechanical properties of the fascia should be highlighted in the discussion of study limitations. Also inability to concurrently observe fibre orientation changes is one of the limitations (DOI:10.21062/ujep/192.2018/a/1213-2489/MT/18/5/866).

 Minor comments

Line 3: Correct cadavers to cadaver

Lines 22-24: Rephrase the aim of the study for clarity

Lines 29-30:  towards the management of a personalized medicine. Rewrite for clarity.

Line 34: Aponeurotic fascia is a type of deep fascia and not a synonym of deep fascia.

Lines 36-37: It is confusing and incorrect to refer to the multi-layered organisation of the fascia lata, whereas it typically has 2 layers.

Line 189: Provide more details about the observed tissue variability among the donors.

Author Response

We have addressed the reviewer's comments in the attached file. Thank you
